# Reconstructing Signals in Millimeter Wave Channels Using Bayesian-Based Fading Models

Claudio Bastos Silva *, Pedro E. Pompilio , Theoma S. Otobo  and Horacio Tertuliano Filho

Department of Electrical Engineering, Federal University of Parana—UFPR, Av. Cel. Francisco H. dos Santos, 100, Curitiba 81530-000, Brazil; pedropompilio@ufpr.br (P.E.P.); theomaotobo95@gmail.com (T.S.O.); tertuliano@ufpr.br (H.T.F.)
*   Correspondence: clbtx@uol.com.br; Tel.: +55-41-99948-7225

**Abstract:** Fading in communication channels presents eminently stochastic characteristics and is a significant challenge, especially at millimeter wave (mmW) frequencies, where the need for lines of sight and the high attenuation of obstacles complicate transmission. This article presents a model based on Bayesian fundamentals intended to improve the description and simulation of stochastic fading effects in these channels. It also includes the use of signal processing techniques to simulate and reconstruct the received signal, simulating the communication channel with an FIR filter. The results obtained by simulating the model show its ability to efficiently capture rapid and profound variations in the signal, typical of those that occur in urban and suburban environments and transmissions in the mmW spectrum. It also provides greater uniformity in signal reconstruction compared to the traditional models that are in use. Using Bayesian fundamentals, which allow dynamic adaptation to change in channel behavior, can improve the efficiency and reliability of networks, especially modern smart networks. Compared to traditional models, the proposed model offers improved signal reconstruction and fading mitigation accuracy, with prospects for future integration in smart communication systems. The better capacity in signal reconstruction presents itself as a differentiator of the model, suggesting greater precision in data transmission.

**Keywords:** Bayesian model; channel fading; particle filter; signal reconstruction



## 1. Introduction

The development of the telecommunications market is increasingly driving research and development toward intelligent, high-capacity, high-speed wireless networks that can adapt to the environment and provide a wide variety of services, some of which are already in use and others are yet to be discovered.

Building an intelligent environment with devices that interact with the medium and that are capable of connecting to the network at very high speeds will require devices that are aware of their surroundings, learn from environmental variations, and adapt their internal configurations based on new statistical variations.

Fading in a telecommunications channel is a phenomenon that directly affects the quality and stability of the system. It occurs due to variations in the signal amplitude in the channel and has an eminently stochastic behavior. The fading severity depends on factors such as link architecture, propagation scenario, carrier frequency, relative speed (in mobile communications) between transmitter and receiver, and other propagation characteristics of an electromagnetic signal. While traditional models such as those of Rice, Rayleigh, and Nakagami effectively describe fading in the microwave spectrum, there is no agreement on their effectiveness in the millimeter wave (mmW) and submillimeter wave spectrum due to the unique propagation behavior in this frequency range.

Using Bayesian techniques can bring flexibility to modeling the fading phenomenon and signal processing in smart networks, favoring the implementation of AI and machine learning algorithms and improving methods and tactics to neutralize adverse effects

imposed on signal propagation. This ultimately results in more efficient and reliable telecommunications networks.

In the literature, a general Bayesian model was not found to describe the fading phenomenon, only for the analyses of specific characteristics or metrics. Studies [1] investigate fading channel models, comparing traditional models, such as Rayleigh and Nakagami-m, and more advanced models, the κ-μ shadowed model, for mmW frequencies; however, this case unifies a variation in the models with a specific distribution. The authors of [2] present an experimentally validated fading model for Terahertz (THz) wireless communication systems based on ambient measurements. They demonstrate that the α–μ distribution better fits the small-scale fading characteristics in THz channels, outperforming traditional models such as Rice, Nakagami-m, and Rayleigh. These studies use frequentist statistical techniques.

In implementing advanced signal reconstruction techniques, such as particle filters that use Bayesian foundations, it is expected that the models will have better adaptability and performance when applied to Bayesian models or algorithms, such as the one proposed in this work. The work in [3] is relevant, where a multiscale particle filter is employed to extract the channel state information (CSI) from the available noisy observations and obtain sufficient parameter statistics. Based on the dates, the Bayes statistical inference theorem derives a conditional posterior probability density distribution for detection threshold value.

Also, the work of [4] presents an improved particle filtering technique for estimating channel coefficients and detecting signals transmitted over fading channels, especially Rayleigh flat fading. The method improves the accuracy of signal detection in non-Gaussian communication systems.

The authors of [5] deal with improvements in particle filters (PFs) for real-time applications. They propose new Bayesian resampling techniques that increase the efficiency and speed of the filtering process without compromising accuracy.

This paper introduces a model for channel fading in mmW channels based on the Bayesian fundamentals, which, favored by the great increase in computational processing power to deal with large data sets, offers significant advantages over traditional models and can overcome the limitations inherent to each one. Furthermore, the paper demonstrates the effectiveness of the proposed model in mitigating the effects of channel fading, compared with other models, in the reconstruction of the signal at the receiver using signal processing techniques.

## 2. Background

The efficiency and reliability of a communication link depend mainly on the quality of the signal received, which should ideally mirror the transmitted information closely. However, disturbances, like noise and interference generated during transmission, can affect quality. Various factors contribute to these disturbances, such as space loss, thermal noise, echo, co-channel interference, intermodulation, and atmospheric effects. In mobile communications, multipath propagation and the Doppler effect also significantly influence channel behavior [6].

The Doppler effect arises from the motion between the transmitter and receiver, causing changes in frequency due to time compression or expansion, which acts as an additive frequency change over the bandwidth. This effect introduces a frequency shift across the bandwidth, influencing the communication channel's behavior. It can result in changes that either weaken or strengthen signals, leading to a dispersion that widens baseband width usage.

The variation in amplitude of the electromagnetic signal between the transmitting and receiving antennas results in fading, with the severity determined by the factors mentioned earlier, and this phenomenon can occur in the space, time, and frequency domains, particularly affecting mobile communications; consequently, fading has a significant impact on the reliability and stability of mobile communication systems, and while traditional

models adequately describe fading for currently used frequencies, there is no consensus on an effective model for the mmW frequency bands [7].

## 2.1. Propagation in the mmW Spectrum

Signals within the millimeter wave (mmW) spectrum exhibit distinct characteristics, often behaving like optical signals due to their minimal or nonexistent scattering levels; this typically requires a line-of-sight (LOS) link architecture for effective transmission. A limited range, reduced penetration through solid objects, and substantial propagation attenuation characterize these signals. Additionally, they are vulnerable to absorption by vapor molecules, atmospheric gasses, and rain [8]. Even minor angle or path length alterations can lead to spatial fading, disrupting the signal.

Signals traveling through the atmosphere can be absorbed due to the different resonances of oxygen and other gasses. This adds to the natural signal loss that occurs in free space; as a result, certain mmW frequency bands are well suited for very-short-range communications and "whisper radio" applications, where the signal weakens rapidly over distances of only a few meters or even fractions of a meter. Rain and hail can also worsen signal loss, especially at frequencies above 10 GHz. For example, 73 GHz signals can experience up to 10 dB/km of loss during heavy rainfall at a rate of 50 mm/h. Additional antenna or transmission gain can be used to mitigate this signal loss. The signal loss from air-to-ground transmission can also depend on the size and orientation of raindrops and clouds. As a result, connections between satellites or drones may experience a more localized and potentially less severe signal loss due to rain compared to terrestrial connections at mmW frequencies [9,10].

Building penetration presents a major challenge, setting it apart from current UHF/microwave systems. The effect of different materials on signal strength is significant. Tests conducted at 38 GHz showed a signal loss of almost 25 dB for a stained glass window and 37 dB for a glass door. Similarly, tests at 28 GHz revealed signal losses of 40.1 dB for external colored glass and 28.3 dB for brick pillars. In contrast, internal clear glass and drywall showed losses of only 3.6 dB and 6.8 dB, respectively [9,11].

Transmission in the mmW spectrum must quickly explore and adapt to the spatial dynamics of the wireless channel. Diffuse scattering on rough surfaces can introduce significant signal variations over very short distances. This necessitates anticipating these rapid variations to properly design channel state feedback, link adaptation, and beamforming/tracking algorithms. Furthermore, near-coincident multipath due to wavelength can create severe small-scale variations in the channel frequency response [9,12].

## 2.2. Fading Models

In the current UHF/microwave frequency spectrum, models built based on the principles of frequentist or classical statistics efficiently elucidate the fading phenomenon. However, these models, each with their particularities, are not general and have certain restrictions or limitations in their practical application. These limitations arise from the mathematical formulation, which may not be suitable for a given link architecture or may not meet the physical properties of the signal propagation behavior, notably influenced by the carrier frequency. These factors impose significant constraints on the effectiveness of the models. Therefore, reaching a consensus on an effective model for the mmW frequency range [8] remains a challenge.

This work proposes and analyzes a fading model based on Bayesian fundamentals and will briefly show the fading models' bases that are in current use—Rician, Rayleigh, and Nakagami-m. The analysis can be further extended to encompass other models and distributions $(\alpha - \eta - k - \mu)$, which provide a unified representation of the Nakagami-m, Rician, and Rayleigh models. These models and variants are based on frequentist (parametric) statistics and are typically associated with nonlinearity and communication in non-line-of-sight (NLOS) scenarios [13,14]. In contrast, within the millimeter wave (mmW) spectrum, the link architecture operates predominantly under line-of-sight (LOS) conditions.

### 2.2.1. Rician Model

Rice's model is expressed by the probability density function (PDF) given by:

$$PDF(p) = \frac{p}{\sigma^2} e^{-\left(\frac{p^2 + A^2}{2\sigma^2}\right)} . I_0\left(\frac{A.p}{\sigma^2}\right) \tag{1}$$

where A is the maximum voltage or power envelope, σ is the standard deviation, and $I_0$ is the modified Bessel function of the first kind and order zero.

The Rician K factor estimates the ratio between the direct signal (LOS) and the contribution of the multipath components that reach the receiver.

$$K = \frac{LOS}{MULTIPATH} \frac{[POWER]}{[POWER]} = \frac{A^2}{2\sigma^2} \tag{2}$$

Rice's PDF can be written as follows:

$$PDF(p) = \frac{p}{\sigma^2} e^{-\frac{p^2}{2\sigma^2}} . e^{-K} I_0\left(\frac{p}{\sigma}\sqrt{2k}\right) \tag{3}$$

By analyzing the PDF expression, it is observed that critical situations occur when:

- $K = 0$

In this condition, there is an NLOS signal only, and the term $e^{-K} = 1$ e $I_0(0) = 1$.

- $K \rightarrow \infty$

If there is a reasonable clearance (LOS) between the transmitter and the contribution of multipath's components is irrelevant, in that case, a fact that increases with the increase in frequency, then K tends to infinity (K→∞) and Rice's PDF tends to the Dirac delta shaped [15].

When analyzing the Rice model's probability density function (PDF) expression, it becomes evident that this model is unsuitable for links operating at high frequencies, such as those in the mmW spectrum. The significant loss due to free space attenuation necessitates a high density of antennas, which almost always require a line-of-sight (LOS) architecture. Additionally, the signal spreading effect is either weak or nonexistent, resulting in a minimal contribution from multipath components.

### 2.2.2. Rayleigh Model

In densely populated urban and suburban areas, it is uncommon for a line-of-sight (LOS) signal to reach the receiver; instead, the received signal is primarily the result of scattering and multipathing. The received signal is a combination of the sum of these scattered and multipath signals, which vary in amplitude and phase, leading to constructive or destructive interference. As a result, the signal tends to fluctuate rapidly and significantly, causing fading.

This is the case where in Rice's PDF expression, $K$ tends to zero (K→0) and the term $e^{-K} = 1$ e $I_0(0) = 1$, resulting in the following:

$$PDF(p) = \frac{p}{\sigma^2} e^{-\frac{p^2}{2\sigma^2}} \tag{4}$$

This expression represents the Rayleigh model's probability density function (PDF). Consequently, the Rayleigh model can be considered a particular case derived from the Rician model when there is no line-of-sight (NLOS) between the transmitter and receiver.

### 2.2.3. Nakagami-m Model

The Nakagami model's probability density function (PDF) represents a continuous probability distribution designed to address the limitations of the Rice K factor. This

model is based on the Gamma distribution and belongs to the family of long-tail distributions, making it particularly effective in modeling fading environments compared to those described by the Rayleigh and Rician models. The Nakagami model can accommodate a wider range of fading scenarios by offering greater flexibility in fitting empirical data, providing a more accurate representation of signal behavior in diverse propagation environments [15,16].

$$PDF(p) = \frac{2m^m}{\Gamma(m)\Omega^m} y^{2m-1} e^{-\frac{m}{\Omega}y^2} \tag{5}$$

where $\Omega$ is the scale control parameter ($\Omega > 0$); $m$ is the shape parameter ($m > 0.5$), which determines the depth of fading and the distribution's tail extension; and $y$ is the random variable that represents the signal's amplitude.

The Nakagami distribution exhibits specific characteristics depending on its mmm parameter. As $m$ tends to infinity, the channel behaves like a static channel, indicating the absence of fading. When $m$ approaches 1, the Nakagami distribution converges to the Rayleigh model. For values of $m$ greater than 1 ($m > 1$), the Nakagami distribution increasingly resembles the Rice model.

## 3. Bayesian Fundamentals

The Bayesian approach is a powerful tool for statistical inference and advanced machine learning models. It helps calculate the probability of specific situations by using available information and conditional probabilities based on their correlation with known scenarios. This method provides a more flexible and intuitive approach to statistical modeling than traditional methods. By incorporating prior knowledge and updating it with new data, Bayesian inference can handle complex data structures and offer more accurate predictions [17]. Additionally, Bayesian models allow the integration of expert knowledge, reducing uncertainty and enhancing model performance.

By logically updating hypotheses, it is possible to explain the probability of an event based on previous data, resulting in a clearer and more accurate understanding of the event's likelihood. This approach allows for more confident decision-making, offering a more intuitive and understandable interpretation of statistical results [18].

*Bayes' Theorem*

Bayes' theorem requires a joint probability distribution for estimating an unknown parameter A from known data B. This joint probability density function can be obtained by multiplying the distribution of A with the data distribution P(B | A). The resulting joint distribution can then be used to estimate the posterior probability distribution of A given B.

The Bayes' theorem expression is given by:

$$p(A|B) = \frac{P(B|A) * P(A)}{P(B)} \tag{6}$$

where $P(B) = \sum_A P(B|A). P(A)$ is the sum over all possible values of A, or in the case of a continuous value of $A$, $P(B) = \int p(B|A).p(A)dA$. The probability of A before considering $B$ is the prior probability ($P(A)$), while the probability of $A$ given $B$ ($P(A|B)$) is called the posterior probability. $P(B | A)$ is the likelihood function that despite having the same meaning as probability, brings a subtle difference to the statistical analysis, describing the probability of observing data already found; it refers to past events with known outcomes [19].

## 4. Proposed Model

A practical approach to determining the a priori distribution involves simulating a sample of known data, when possible, and examining the resulting random variable distribution. Several graphs can be created based on the chosen distribution, including

a Probability Distribution Function (PDF), Cumulative Distribution Function (CDF), histogram, and QQplot.

*Choosing Model Distributions*

Simulating a signal transmitted with a 38.0 GHz carrier, BPSK modulation (to simplify analysis), and considering the effects of free space attenuation (500 m), white noise (AWGN), atmospheric noise, vapor absorption, and the Doppler effect for the receiver with a relative speed of 60 km/h, the resulting distribution of noise is shown in the histogram and QQ plot presented in Figure 1.

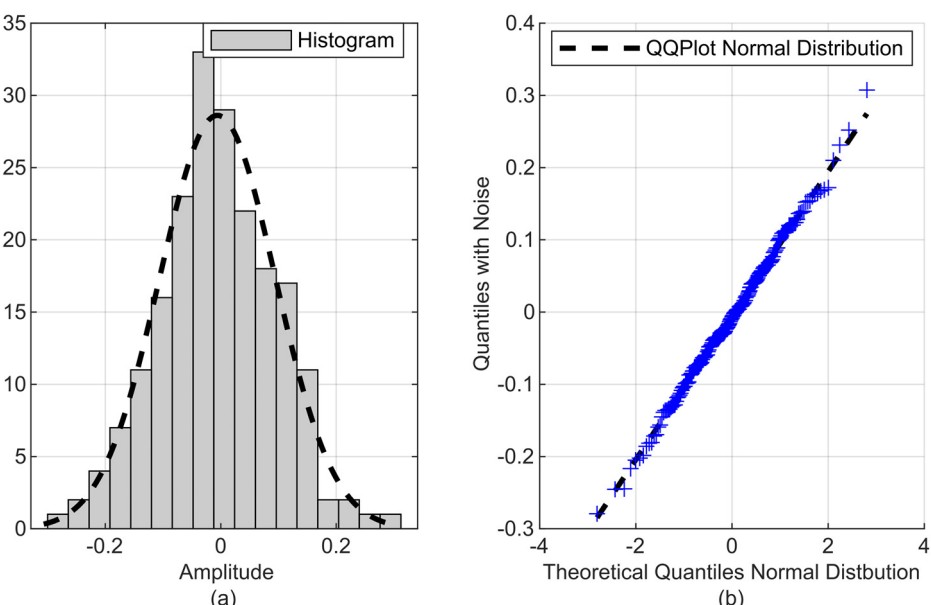

**Figure 1.** (**a**) Noise distribution histogram and (**b**) QQplot for normal distribution.

The simulation graphs (Figure 1) show that the noise distribution in the receiver adheres to the normal distribution. Therefore, this will be chosen as the a priori distribution.

The Gamma distribution, which includes shape ($\alpha$) and scale control parameters ($\theta$) in its formulation, offers flexibility to find the best fit. These parameters allow the model to adapt to a wide range of data patterns, making it a robust choice for modeling diverse scenarios. Furthermore, the Gamma distribution is the basis of the well-established Nakagami-m model, widely used to describe channel fading. Given its versatility and mathematical properties, this study will use the Gamma distribution as a posterior distribution.

Substituting into the expression of Bayes' theorem (6) and simplifying, we obtain the following:

$$p(X|Y) = \frac{Y^{\alpha-1}.e^{-\frac{Y}{\theta}}.e^{-\frac{(X-\mu)^2}{2\sigma^2}}}{\int Y^{\alpha-1}.e^{-\frac{Y}{\theta}}.e^{-\frac{(X-\mu)^2}{2\sigma^2}} \, dX} \tag{7}$$

The denominator is a normalization integral representing the sum of all mutually exclusive hypotheses.

$$\int Y^{\alpha-1}.e^{-\frac{Y}{\theta}}.e^{-\frac{(X-\mu)^2}{2\sigma^2}} \, dX \tag{8}$$

Solving for $X$:

$$\int e^{-\frac{(X-\mu)^2}{2\sigma^2}} dX =$$
$$u = \frac{X-\mu}{\sigma} \quad du = \frac{dX}{\sigma} \quad dX = \sigma du$$

Substituting:

$$\int e^{-\frac{(X-\mu)^2}{2\sigma^2}} dX = \int e^{-\frac{u^2}{2}} \sigma du$$

This is the integral of the standard Gaussian function; its domain extends over the real domain $(-\infty, \infty)$. Using the general form:

$$I = \int_{-\infty}^{\infty} e^{-ax^2} dx$$
$$I^2 = \left(\int_{-\infty}^{\infty} e^{-ax^2} dx\right).\left(\int_{-\infty}^{\infty} e^{-ay^2} dy\right)$$
$$I^2 = \int_{-\infty}^{\infty} \int_{-\infty}^{\infty} e^{-a(x^2+y^2)} dx dy$$

$x^2 + y^2$ is the equation of a circle of radius $r^2$. Transforming to polar coordinates:

$$x^2 + y^2 = r^2$$
$$dx dy = r dr d\theta$$

With r varying from 0 to $\infty$ and $\theta$ from 0 to $2\pi$:

$$I^2 = \int_0^{2\pi} d\theta \int_0^{\infty} e^{-ar^2}.r dr$$

Solving for $r$:

$$u = ar^2 \implies du = 2ar dr$$
$$\int_0^{\infty} e^{-ar^2}.r dr = \frac{1}{2a} \int_0^{\infty} e^{-u} du = \frac{1}{2a}$$

Integrating with respect to $\theta$:

$$I^2 = \frac{1}{2a} \int_0^{2\pi} d\theta = \frac{\pi}{a}$$
$$I^2 = \frac{\pi}{a} \implies I = \sqrt{\frac{\pi}{a}} \tag{9}$$

In this case, $a = \frac{1}{2}$.

Returning the original variable X and replacing, we obtain the following:

$$\int e^{-\frac{(X-\mu)^2}{2\sigma^2}} dX = \sigma\sqrt{2\pi} \tag{10}$$

Integrating with respect to $Y$:

$$\int Y^{\alpha-1}.e^{-\frac{Y}{\theta}}.dY$$

This integral can be solved using rules applicable to the Gamma function.

$$u = \frac{Y}{\theta} \quad Y = \theta u \quad \text{e} \quad dY = \theta du$$
$$\int Y^{\alpha-1}.e^{-\frac{Y}{\theta}} dY = \int (\theta u)^{\alpha-1}.e^{-u}.\theta du$$

simplifying:

$$\int Y^{\alpha-1}.e^{-\frac{Y}{\theta}} dY = \theta^{\alpha} \int u^{\alpha-1}.e^{-u} du$$

The integrand is the definition of the Gamma function.

$$\theta^{\alpha} \int u^{\alpha-1}.e^{-u} du = \theta^{\alpha}.\Gamma(\alpha) \tag{11}$$

Finally, the solution for (8) is then:

$$\int Y^{\alpha-1}.e^{-\frac{Y}{\theta}}.e^{-\frac{(X-\mu)^2}{2\sigma^2}} dX = \sigma\sqrt{2\pi}.\theta^{\alpha}.\Gamma(\alpha) \tag{12}$$

Substituting in (8):

$$p(X|Y) = \frac{Y^{\alpha-1}.e^{-\frac{Y}{\theta}}.e^{-\frac{(X-\mu)^2}{2\sigma^2}}}{\sigma\sqrt{2\pi}.\theta^{\alpha}.\Gamma(\alpha)} \tag{13}$$

$$\mathcal{L}(Y,\alpha,\theta) = \frac{\sqrt{2\pi}}{2\pi\sigma\theta^{\alpha}\Gamma(\alpha)}.Y^{\alpha-1}e^{-(\frac{Y}{\theta}+\frac{(X-\mu)^2}{2\sigma^2})} \tag{14}$$

With $Y > 0$ and the other parameters following the definitions of the original Gamma and normal distributions.

The curves in Figure 2 depict how the probability density function of the $\mathcal{L}$ distribution changes with variations in $Y$.

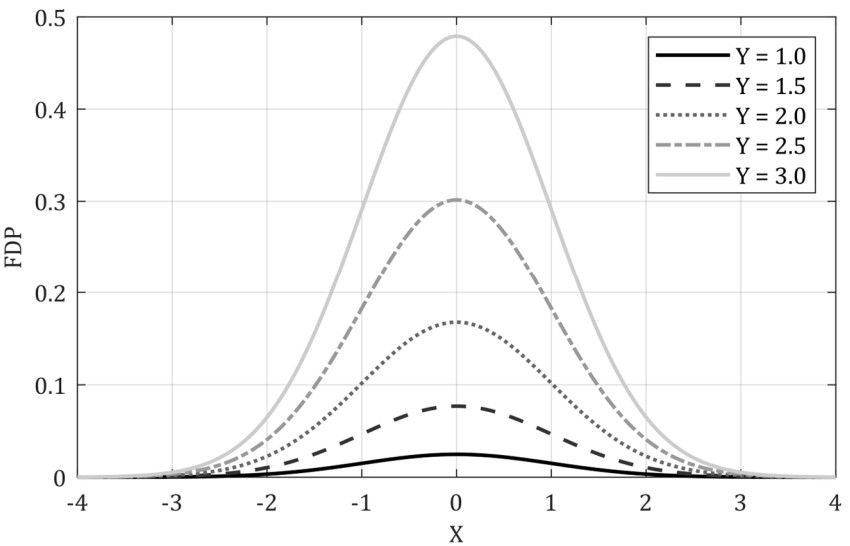

**Figure 2.** $\mathcal{L}$ distribution's behavior as a function of the variation in $Y$.

The variable being considered in the posterior distribution is $Y$. To calculate the probability density function (PDF), each term in the prior distribution, which is a normal distribution, is multiplied by the corresponding term in the posterior distribution. Changing the $Y$ value helps evaluate how the posterior distribution affects the previous distribution and how one is affected by the other.

In the posterior distribution, the variable $Y$ can represent any parameter, whether a function, vector, or distribution. This variable can encompass a physical quantity or a parameter that influences propagation, such as the power of a received signal. $Y$'s flexibility allows for a comprehensive analysis of various factors that impact signal behavior.

## 5. Simulations and Results

The objective of the simulation is to show, in a comparative manner, the behavior of the fading channel for the proposed $\mathcal{L}$ and Nakagami models. The Rice and Rayleigh models will not be included in the comparisons because they do not adequately represent the propagation model in the mmW spectrum due to the links (LOS) architecture or the low or nonexistent spreading of the signal in this spectrum band.

The simulation parameters follow those previously established, namely "a signal transmitted with a 38.0 GHz carrier, BPSK modulation (to simplify analysis), and considering the effects of free space attenuation (500 m), white noise (AWGN), atmospheric noise, vapor absorption, and Doppler effect for the receiver with a relative speed of 60 km/h".

Also, using an FIR filter (Finite Impulse Response) with a bandwidth of 100 MHz [20] to simulate the channel, the signal will be reconstructed for each model to compare the efficiency of mitigating the effects of channel fading.

The choice of the FIR filter is due to its fundamental characteristics [17].

- Guaranteed stability—its coefficients depend only on the input samples.
- Linear phase—they can be implemented to have a linear phase response; they do not distort the signal waveform.
- Easy implementation on digital hardware as it does not require feedback.
- Flexibility in design—can be easily adjusted to meet specific filter requirements, such as low pass, high pass, and bandwidth.

A typical phase sample of a transmitted signal with the effect of noise, based on (5) and (14), is shown in Figure 3.

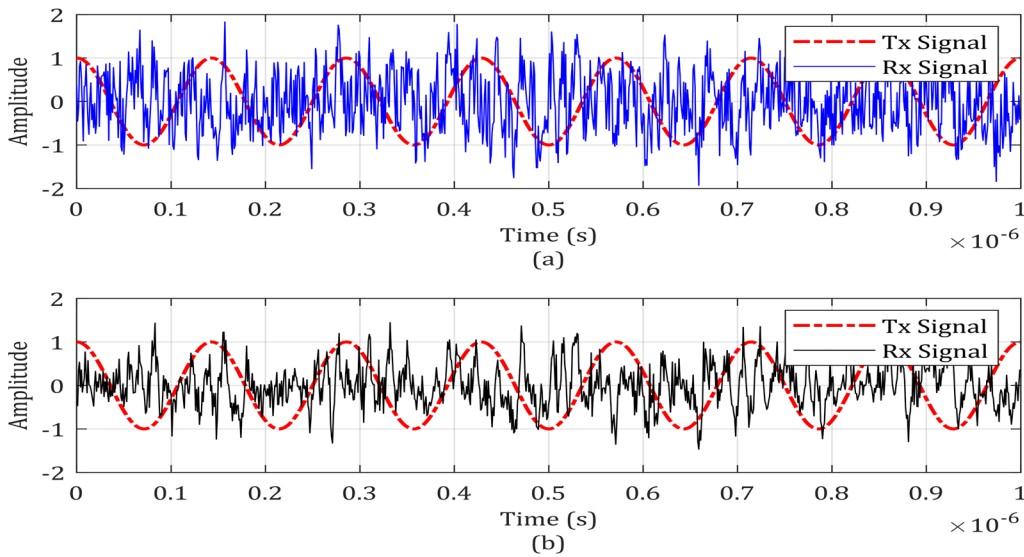

**Figure 3.** Pure sinusoidal and received signal with noise. (**a**) $\mathcal{L}$ model; (**b**) Nakagami.

Figure 3 shows the noise generation in the channel for the models. The proposed $\mathcal{L}$ model captures a more significant number and amplitude of the noise generated in the transmission. In other words, it is more sensitive to rapid and profound signal variations typical of those in dense urban and suburban environments.

The scatterplot in Figure 4 shows the noise amplitude distribution for both models over the sampled period.

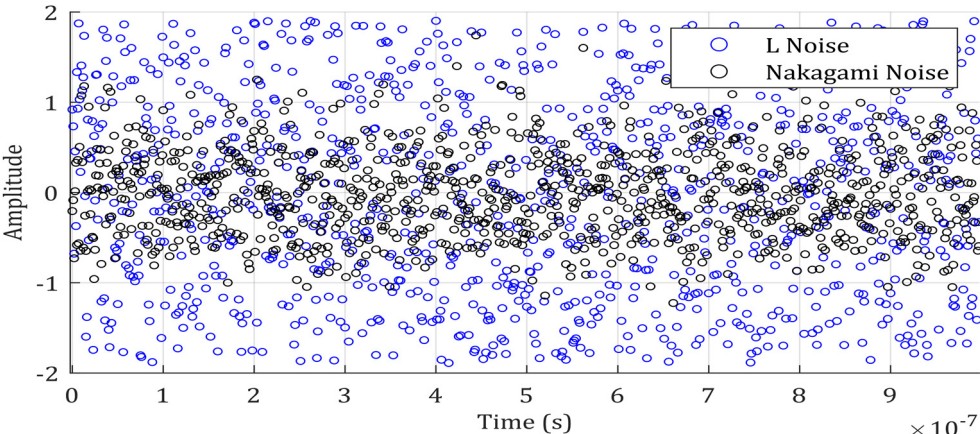

**Figure 4.** Compared dispersion of amplitude distribution.

From the probability distribution of the $\mathcal{L}$ model (14), the Nakagami model (5), and the response of the FIR filter, it is possible to determine the impulse response of the communication channel from the perspective of the fading effect for each model.

Impulse response describes how the channel affects a transmitted signal, especially in terms of signal distortion and dispersion over time; it shows how the channel changes an impulse (a very short signal, ideally of infinitesimal width).

For visualization purposes, in Figure 5, only 100 samples are shown.

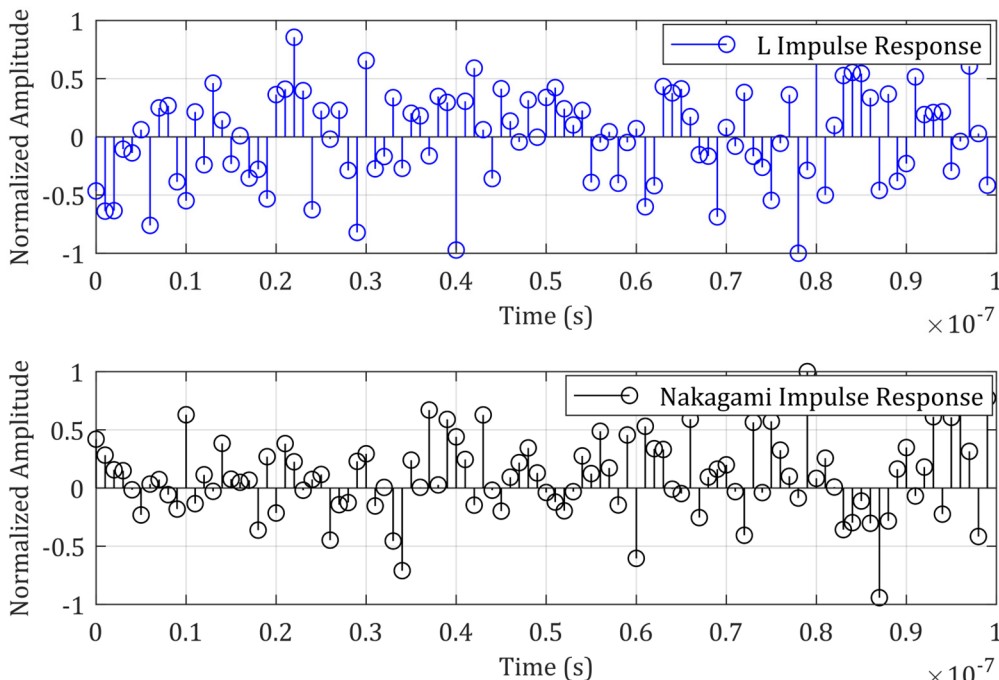

**Figure 5.** Channel unitary impulse response simulated for a sample in the period.

Taking the impulse response, using the fast Fourier transform (*FFT*) and the inverse Fourier transform (*IFFT*), and using signal processing techniques, the signal can be reconstructed.

The signal reconstruction adopts the following procedure, using the fast Fourier transform (*FFT*) to compute the Fourier transforms and the inverse fast Fourier transform (*IFFT*) to return to the time domain. Given $h(t)$, one can calculate $H(f)$ using the *FFT*:

$$H(f) = FFT(h(t)) \tag{15}$$

To determine the *FFT* of the input signal $x(t)$, one can calculate $X(f)$ using the *FFT* algorithm.

$$X(f) = FFT(x(t)) \tag{16}$$

To obtain $Y(f)$, multiply $X(f)$ by $H(f)$, $[Y(f) = X(f).H(f)]$, and apply *IFFT* to $Y(f)$ to obtain $y(t)$:

$$y(t) = IFFT(Y(f)) \tag{17}$$

Using this procedure (*FFT* and *IFFT*), the convolution process is simplified by transforming it to the frequency domain, where the convolution becomes a simple multiplication; this is computationally efficient and practical to implement.

Figure 6 shows a comparative reconstruction of the signal shown in Figure 4. For the proposed $\mathcal{L}$ model, 3000 iterations were performed (in Figure 6) to refine the reconstruction process, and this is one of the advantages of the proposed model, which, being built on Bayesian fundamentals, allows for this refinement through iterative processes.

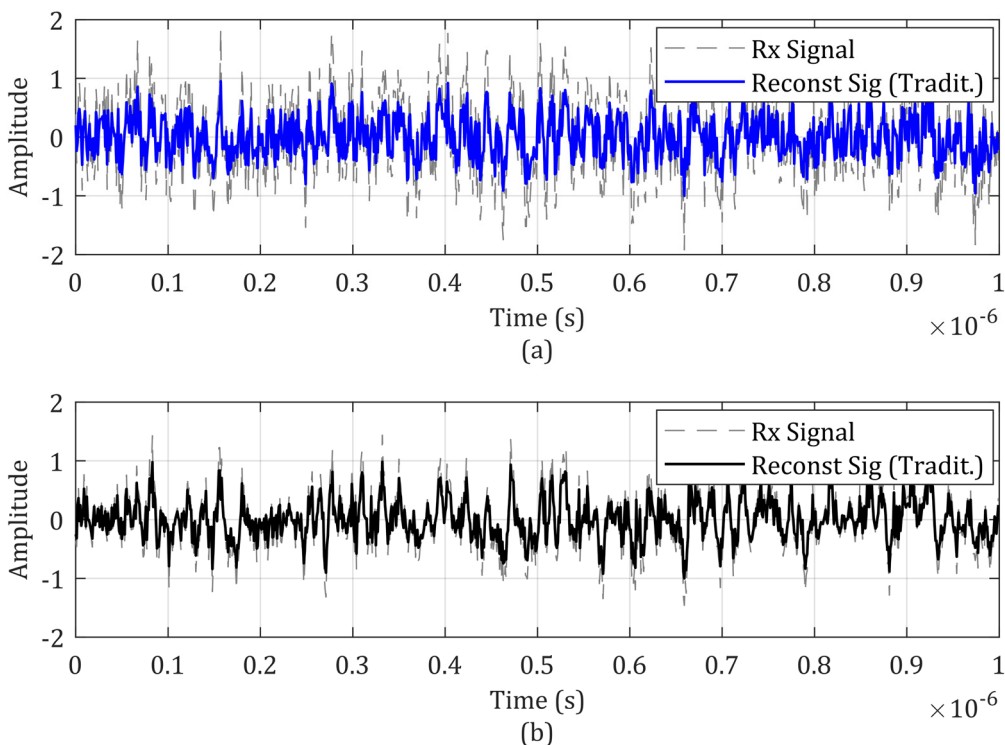

**Figure 6.** Reconstructed signal: (**a**) $\mathcal{L}$ model; (**b**) Nakagami.

It can be seen that the $\mathcal{L}$ model is, by construction, less conservative than the Nakagami model (Figures 4 and 5); that is, it has greater flexibility by allowing the signal to oscillate at larger amplitudes, capturing possible variations that can cause the fading effect. However, it has a greater response uniformity when reconstructed than the compared model.

The analysis of the "Comparative PDFs of reconstructed signal noise" (Figure 7) shows that the Nakagami distribution presents a smaller dispersion of noise values (narrower base); also, the higher probability density in the center (height of the curve) implies that most of the noise values for the reconstructed signal are concentrated near the mean value.

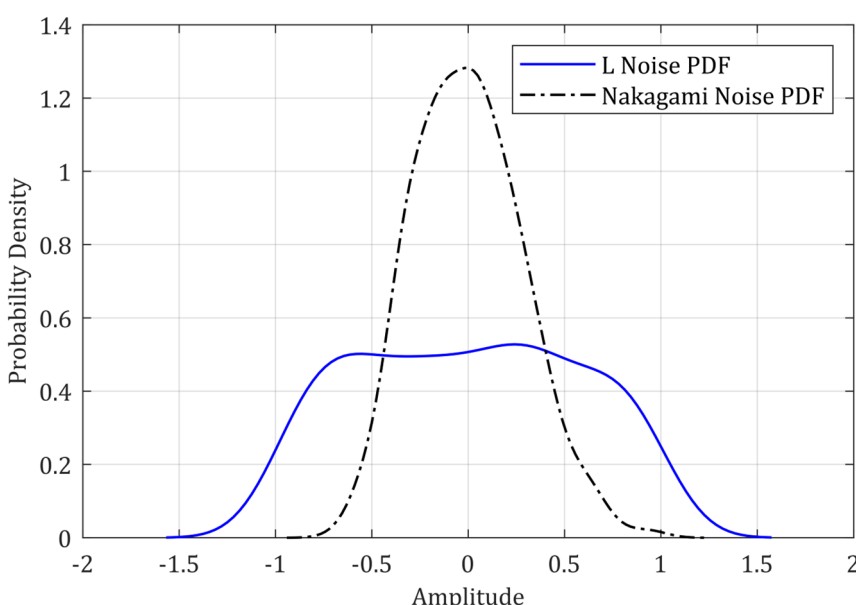

**Figure 7.** Comparative PDFs of reconstructed signal noise.

The broader base of the $\mathcal{L}$ distribution shows a greater noise dispersion, indicating a wider variation of values around the mean. The lower height of the probability density curve in the center implies that the values are less concentrated and more distributed.

The Nakagami model, due to its lower noise dispersion, tends to result in a more stable and predictable signal, which makes it advantageous in environments where robustness against signal variations is critical. As a disadvantage, it has less capacity to adapt to rapid and profound channel variations, making it less efficient in communication environments with many fluctuations.

The proposed $\mathcal{L}$ model, in turn, has a greater capacity to represent channel variations and fluctuations, such as those expected in propagation in the mmW spectrum, and can better adapt to rapid and significant changes in the communication environment. As a disadvantage, the greater dispersion of noise values can result in a less stable signal that is more susceptible to interference.

In summary, with its greater dispersion, the $\mathcal{L}$ model can represent a wider variety of noise scenarios, which is advantageous in highly variable communication environments. The ability to handle severe fluctuations makes the model adaptable to different channel conditions, which provides robustness in unpredictable environments. The model is, therefore, advantageous in scenarios where adaptation to significant variations in noise and interference levels is necessary.

Fade channel power (Figure 8) measures the variation in signal strength due to the fading effect in a wireless communication channel. It is calculated by averaging the quadratic values of the received signal. The resulting value represents the signal strength over time, reflecting the variations caused by fading. It is a crucial variable for designing and analyzing wireless communication systems, as it allows for predicting the system performance under different propagation conditions. It helps to implement fading mitigation techniques such as receiver diversity, equalization, and adaptive modulation.

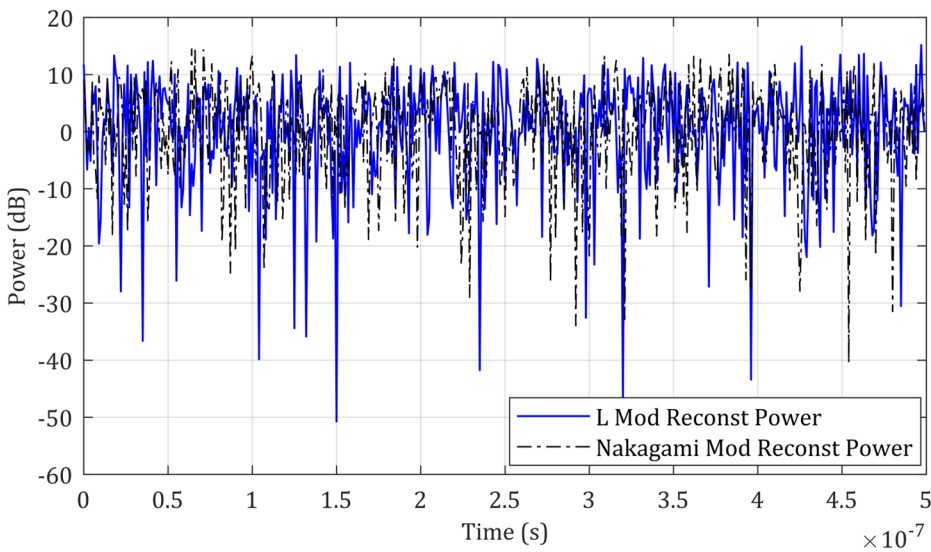

**Figure 8.** Comparative fade channel power of reconstructed signals.

## 6. Using Advanced Techniques

A significant advancement in signal reconstruction techniques can be achieved using particle filters, also known as bootstrap filters or sequential Monte Carlo methods, which are a set of algorithms used to estimate the internal states of a system based on partial and noisy observations. In this filter, Bayesian inference allows for estimating a combined state for a measurement with a prior probability.

Particle filters do not apply directly to the Nakagami model or any other derived from the foundations of classical or frequentist statistics, as these are static models that assume a specific distribution for the magnitude of the fading signal and, by nature, do

not incorporate a dynamic mechanism to update states or incorporate new information as Bayesian methods do.

As it is an algorithm that allows for the continuous updating of the dynamic state of the system, the particle filter can track variables that change over time, such as the communication channel.

*Implementation*

In the implementation of the particle filter, a set of particles (samples) is used to represent the posterior distribution of the state of the system. Each particle has a weight that indicates its probability.

$$\chi = \left\{ \text{ffi} x^{[j]}, \; \omega^{[j]} \text{ffl} \right\}_{j=1, \, ...., \, J} \tag{18}$$

This expression (18) represents the set of particles (samples) that the filter uses to estimate the state of a dynamic system, where $x^{[j]}$ is the jth particle, i.e., a possible sample of the current state or a hypothetical representation of the system's state, and $\omega^{[j]}$ is the weight associated with the jth particle. It represents the relative probability that particle $x^{[j]}$ accurately represents the system's true state. Higher weights indicate that the particle is more in line with current observations.

$$p(x) = \sum_{j=1}^{J} \omega^{[j]} \delta_{x^{[j]}}(x) \tag{19}$$

The more particles that fall into a region, the higher the probability in that region. They are initialized around the received signal with random noise. Each represents a hypothesis about the true state of the signal.

With each iteration, the particles are updated with the addition of noise. The weights are adjusted according to the current observation probability, which measures the similarity between the received signal and the sample.

Resampling is carried out to avoid sample degeneration, where few particles assume significant weight. Resampling according to weight ensures that particles with higher weight (closer to the true signal) are selected more often. The final signal estimate is obtained as the weighted average of the particles.

The particle filter can be especially useful for dealing with uncertainties and rapid signal variations, providing a more accurate estimate of the received signal's state.

Comparatively, the reconstruction of the simulated signal for the $\mathcal{L}$ model using traditional techniques and particle filters is shown in Figure 9:

Bayesian filtering, a natural extension of the proposed $\mathcal{L}$ model, facilitates methodological and conceptual integration, allowing for a cohesive and robust approach to understanding channel fading and signal reconstruction when considering the temporal evolution of the channel state.

Figure 10, which compares the noise powers, shows that the particle filter presents smaller noise variations over time, indicating greater noise suppression efficiency than the traditional method. The particle filter therefore has a greater capacity to attenuate abrupt variations in the resulting noise, resulting in a more stable and accurate signal reconstruction, reducing errors and improving system performance. In practical applications, particle filters can result in a higher quality reconstructed signal, reducing errors and improving the communication system's performan As can be seen in Figure 11 (power Spectral Density).

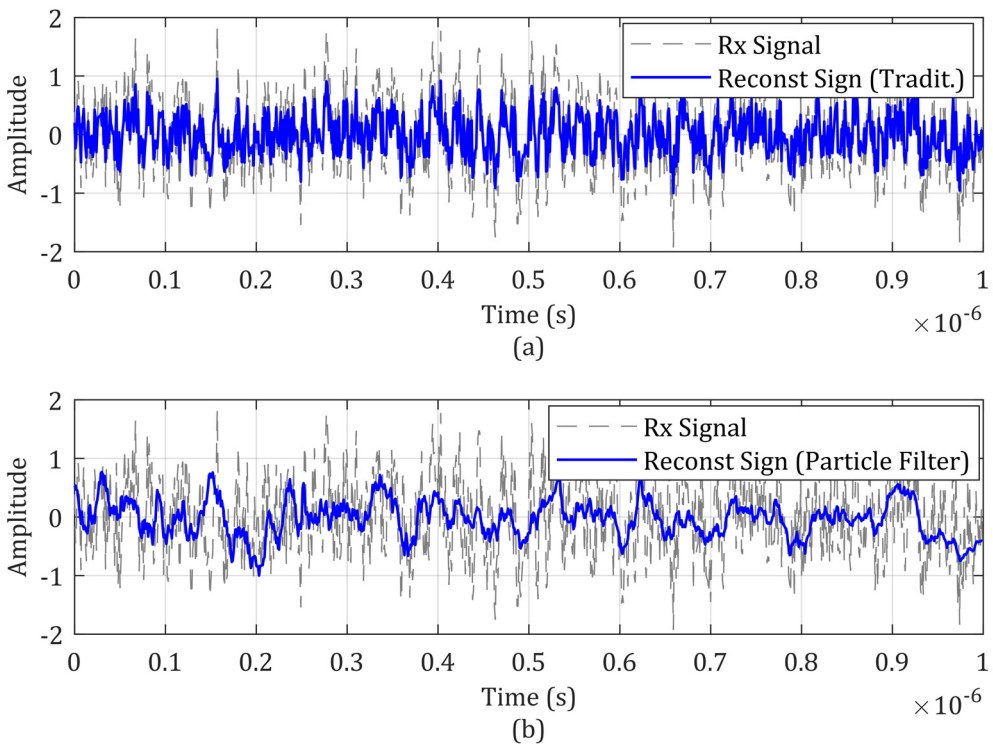

**Figure 9.** Reconstructed signal for the $\mathcal{L}$ model using (**a**) traditional reconstruction and (**b**) particle filter techniques.

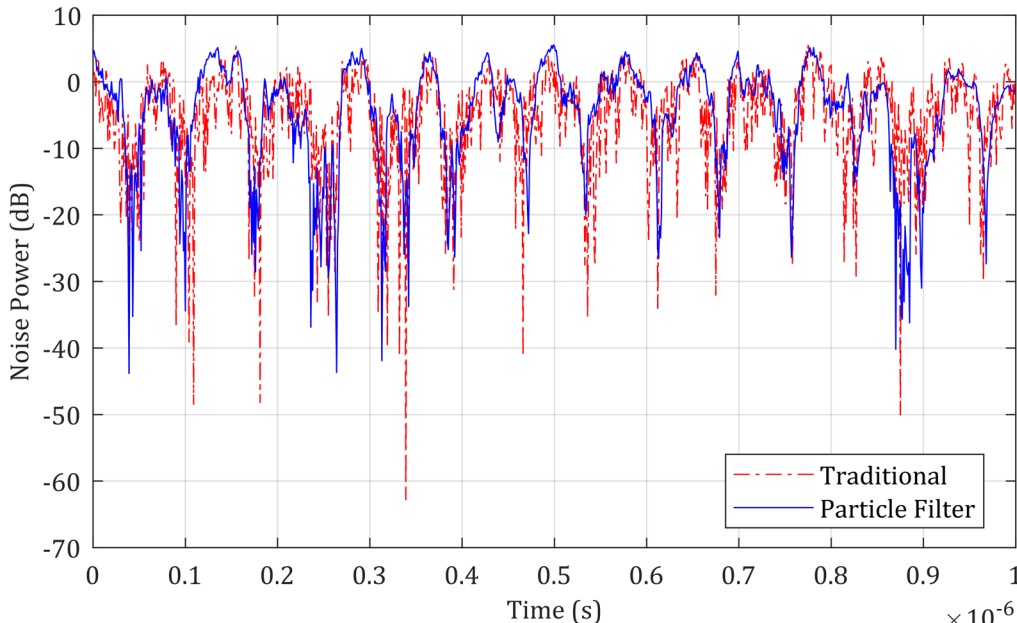

**Figure 10.** Line plot of reconstructed signal noise power in dB ($\mathcal{L}$ model).

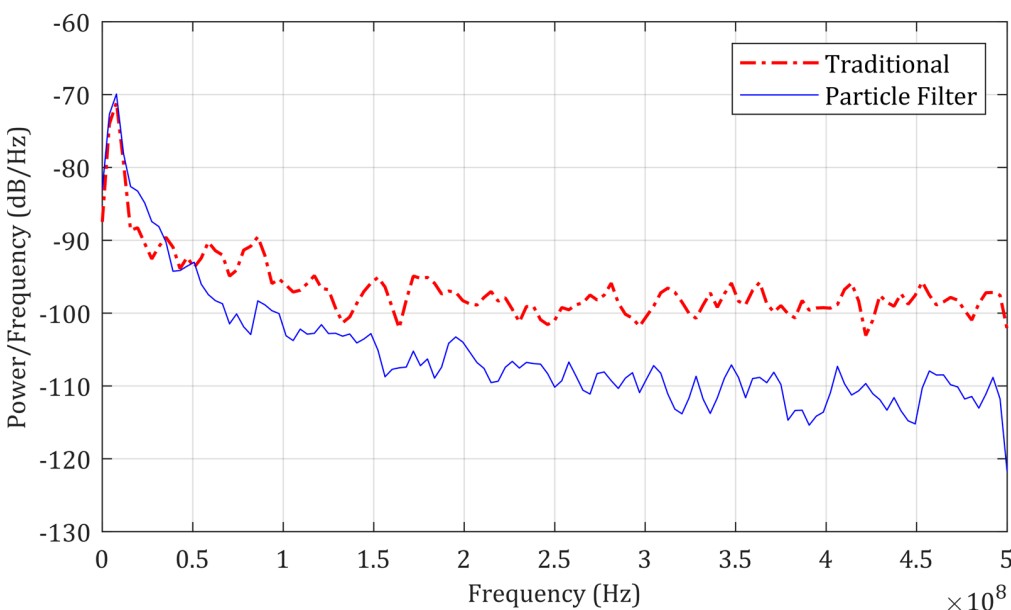

**Figure 11.** Power spectral density of reconstructed signal noise ($\mathcal{L}$ model).

## 7. Conclusions and Future Work

Analysis of the results suggests that the proposed $\mathcal{L}$ model can be a tool for modeling fading in communication channels in the mmW spectrum. Compared to the traditional models (Rice, Rayleigh, and Nakagami), it demonstrates greater flexibility in capturing fast and deep signal variations typical of dense urban and suburban environments and those that occur in millimeter wave propagation. It can represent and describe various fading scenarios, which is particularly important in mmW channels, where the signal is highly influenced by obstacles and attenuations and the architecture of the links, which are almost always a line of sight (LOS).

When using a joint probability distribution, the Bayesian approach allows the model to adapt to channel conditions dynamically, providing greater simulation accuracy and quality in signal reconstruction.

Regarding the signal reconstruction process, the $\mathcal{L}$ model demonstrated an advantage in providing greater uniformity than the Nakagami-m model. This is attributed to its ability to incorporate information about the statistical distribution of fading and specific channel characteristics, resulting in a more accurate reconstruction that is less prone to noise and interference. In addition, the model's iterative structure allows for successive refinements, improving channel estimation accuracy in complex scenarios and directly impacting the reconstructed signal's quality.

In conjunction with the particle filter, the $\mathcal{L}$ model has efficiently captured a wide range of noise behaviors, including fast and deep variations. It quickly adapts to dynamic variations and effectively adjusts the particles and weights in the filter. Noise suppression is more efficient, resulting in reconstructed signals with smaller noise variations and a higher signal-to-noise ratio (SNR).

Overall, the $\mathcal{L}$ model proved a versatile tool for simulating fading in mmW channels and signal reconstruction. These characteristics suggest that the model could be a promising tool for developing more efficient and accurate communication networks with its challenging propagation characteristics in the millimeter wave spectrum. Future developments regarding the model should include validation with real measurements and integration with advanced signal processing techniques, AI algorithms, and machine learning.

**Author Contributions:** Conceptualization, C.B.S., P.E.P. and T.S.O.; Methodology, C.B.S., T.S.O. and H.T.F.; Software, C.B.S. and H.T.F.; Validation, C.B.S. and H.T.F.; Formal analysis, C.B.S., P.E.P. and H.T.F.; Investigation, T.S.O.; Resources, H.T.F.; Writing—original draft, C.B.S.; Visualization, P.E.P.;

Supervision, C.B.S.; Project administration, C.B.S.; Funding acquisition, H.T.F. All authors have read and agreed to the published version of the manuscript.

**Funding:** This work was supported in part by the PPGEE—PROGRAMA DE POS-GRADUAÇÃO_EM ENG. ELETRICA-UFPR Setor de Tecnologia—Centro Politecnico. Av. Cel Francisco H. dos Santos 100—81530-000-Curitiba, PR.

**Data Availability Statement:** Data are contained within the article.

**Conflicts of Interest:** The authors declare no conflicts of interest.

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
