# Peer review of "Reconstructing Signals in Millimeter Wave Channels Using Bayesian-Based Fading Models"

_electronics, doi:10.3390/electronics13224406_

Round 1
Reviewer 1 Report
Comments and Suggestions for Authors
See attached

Reviewer 2 Report
Comments and Suggestions for Authors
This article presents a model based on the Bayesian fundamentals (which also compares with more “traditional” fading models) with the aim to improve the description and simulation of stochastic fading effects in millimeter-wave channels. It also presents signal processing techniques capable of simulating and reconstructing the received signal.
The main novelty of the article lies on the way of improving the description and simulation of stochastic fading effects in millimeter-wave channels (a frequency band where seems to be no consensus regarding the modeling of fading).
The content of the article is within the scope of the Journal.
The literature supports well the article’s case.
Figures support the article’s case.
Conclusions are in accordance with the derived results and provide a hint for future work (validation with real measurements and use of advanced signal processing techniques, AI algorithms, and machine learning.).
Specific comments
I would prefer the term “Bayesian fundamentals” rather than “Bayesian foundations”.
Since Rice and Rayleigh models are not involved in the comparison with the Bayesian one, subsections 2.2.1 and 2.2.2 should only focused on eqs. (1), (2) and (3) and serve as a background for the Nakagami model.
In eq. (2), “k” should be replaced by “K” (capital letter).
The authors should give some explanation about how the eq. (7) was derived from (6) (what substitutions were made?).
The expression in line 249 should not be there (it correctly appears in line 252).
A minus (“-“) is missing in the exponent ar2 (line 265).
“Solving for θ” is not the right phrase (line 269).
The authors should clarify what they mean by “Solving for Y” (line 275) and the integral that follows (line 276).
Though in lines 118-119, the article states that “This work will … compare a proposed model developed on Bayesian foundations with more traditional fading models, Rician, Rayleigh, Nakagami-m …”, the comparison only involves the Nakagami model (lines 310-314).
In eq. (17), Y(t) should be replaced by y(t).
The text in lines 350–366 and 369–391 should be considerably shortened and only focus on eq. in line 378 and eq. (20). The rest of the text is useful but rather trivial for those involved in the field.
Use of English
The article is generally well written, so regarding the use of English a minor editing would be sufficient.
Review decision
I consider the article publishable subject to the proposed revisions.
Comments on the Quality of English LanguageUse of English
The article is generally well written, so regarding the use of English a minor editing would be sufficient.
Reviewer 3 Report
Comments and Suggestions for Authors
This article presents a model based on Bayesian fundamentals that allows allow dynamic adaptation to changes in channel behavior, can improve the efficiency and reliability of networks, especially modern smart networks. The proposed Bayesian model introduces a flexible and dynamic alternative to traditional fading models like Rayleigh, Rician, and Nakagami. It also includes the use of signal processing techniques to simulate and reconstruct the received signal, simulating the communication channel with an FIR filter.
Present the performance improvement of the proposed model compared to other existing fading models in the abstract section and the future prospects.
Comparison with relevant existing research works is missing which is important for better illustration pointing the strength of present study.
Present the simulation parameters used in the results in a tabular form.
What is the computational complexity, accuracy and real-time applicability of the model contemplating it in practical systems.
Round 2
Reviewer 2 Report
Comments and Suggestions for Authors
I am happy with the authors response including their rational for keeping 2.2.1 and 2.2.2 as separate sub-sections.
My only remark regards lines 154-155 (of the revised text which needs rephrasing).
Given the above, I consider the article publishable practically as it is (with a minor editing regarding the use of English).
Comments on the Quality of English LanguageRegarding the use of English, a minor editing would be sufficient.
Author Response
See Attached:

Reviewer 3 Report
Comments and Suggestions for Authors
Authors have successfully address the comments. The manuscript can be accepted.
Author Response
See attached:
